# Association of Glutathione Peroxidase 3 (GPx3) and miR-196a with Carbohydrate Metabolism Disorders in the Elderly

**DOI:** 10.3390/ijms25105409

**Published:** 2024-05-15

**Authors:** Adam Włodarski, Izabela Szymczak-Pajor, Jacek Kasznicki, Egle Morta Antanaviciute, Bożena Szymańska, Agnieszka Śliwińska

**Affiliations:** 1Department of Nucleic Acid Biochemistry, Medical University of Lodz, 92-213 Lodz, Poland; adam.wlodarski@stud.umed.lodz.pl (A.W.); izabela.szymczak@umed.lodz.pl (I.S.-P.); 2Department of Internal Diseases, Diabetology and Clinical Pharmacology, Medical University of Lodz, 92-213 Lodz, Poland; jacek.kasznicki@umed.lodz.pl; 3Centre for Cellular Microenvironments, Mazumdar-Shaw Advanced Research Centre, University of Glasgow, Glasgow G12 8QQ, UK; 2145539A@student.gla.ac.uk; 4Research Laboratory CoreLab, Medical University of Lodz, Mazowiecka 6/8 St., 92-215 Lodz, Poland; bozena.szymanska@umed.lodz.pl

**Keywords:** microRNA-196a, GPx3, prediabetes, oxidative stress, biomarker, obesity, insulin resistance, diabetes, hyperglycemia

## Abstract

The escalating prevalence of carbohydrate metabolism disorders (CMDs) prompts the need for early diagnosis and effective markers for their prediction. Hyperglycemia, the primary indicator of CMDs including prediabetes and type 2 diabetes mellitus (T2DM), leads to overproduction of reactive oxygen species (ROS) and oxidative stress (OxS). This condition, resulting from chronic hyperglycemia and insufficient antioxidant defense, causes damage to biomolecules, triggering diabetes complications. Additionally, aging itself can serve as a source of OxS due to the weakening of antioxidant defense mechanisms. Notably, previous research indicates that miR-196a, by downregulating glutathione peroxidase 3 (GPx3), contributes to insulin resistance (IR). Additionally, a GPx3 decrease is observed in overweight/obese and insulin-resistant individuals and in the elderly population. This study investigates plasma GPx3 levels and miR-196a expression as potential CMD risk indicators. We used ELISA to measure GPx3 and qRT-PCR for miR-196a expression, supplemented by multivariate linear regression and receiver operating characteristic (ROC) analysis. Our findings included a significant GPx3 reduction in the CMD patients (n = 126), especially in the T2DM patients (n = 51), and a decreasing trend in the prediabetes group (n = 37). miR-196a expression, although higher in the CMD and T2DM groups than in the controls, was not statistically significant, potentially due to the small sample size. In the individuals with CMD, GPx3 levels exhibited a negative correlation with the mass of adipose tissue, muscle, and total body water, while miR-196a positively correlated with fat mass. In the CMD group, the analysis revealed a weak negative correlation between glucose and GPx3 levels. ROC analysis indicated a 5.2-fold increased CMD risk with GPx3 below 419.501 ng/mL. Logistic regression suggested that each 100 ng/mL GPx3 increase corresponded to a roughly 20% lower CMD risk (OR = 0.998; 95% CI: 0.996–0.999; *p* = 0.031). These results support the potential of GPx3 as a biomarker for CMD, particularly in T2DM, and the lack of a significant decline in GPx3 levels in prediabetic individuals suggests that it may not serve reliably as an early indicator of CMDs, warranting further large-scale validation.

## 1. Introduction

Over the preceding few decades, the worldwide escalation in carbohydrate metabolism disorders (CMDs) encompassing prediabetes and type 2 diabetes mellitus (T2DM) has been remarkable, driven partly by urbanization and lifestyle changes [1]. CMDs, prevalent especially in those above 65 years of age, encompass a spectrum of metabolic abnormalities, correlating with conditions such as insulin resistance (IR), obesity, and hypertension [2,3]. The management of CMDs presents significant economic and logistical burdens to healthcare systems. Current estimates suggest a substantial projected increase in the prevalence of T2DM and prediabetes worldwide, emphasizing an urgent need for effective strategies to address this public health challenge, particularly among the elderly population [2]. In the United States, about 25% of adults aged 65 and older are diagnosed with diabetes, while over half may be classified as prediabetic [4].

First, since 1956, the oxidative theory of aging has become influential [5]. Age-related oxidative stress (OxS) arises from increased free radical production, reduced antioxidants, impaired antioxidant enzymes, and compromised repair mechanisms [6]. Secondly, CMDs form a mutually reinforcing vicious circle. IR results in insufficient cellular response to insulin, causing hyperglycemia (HG). Both high glucose and insulin levels exacerbate oxidative stress (OxS), amplified by aging [6,7,8]. HI and HG boost nicotinamide adenine dinucleotide phosphate (NADPH) oxidase activity, and excessive mitochondrial glucose metabolism triggers an augmented electron flux through the electron transport chain, resulting in a surge in superoxide anion production, which accelerates other reactive oxygen species (ROS) generation [9,10,11,12]. These ROS disrupt insulin signaling via the phosphatidylinositol 3-kinase (PI3K/Akt) pathway, induce pancreatic β-cell dysfunction, and activate inflammatory pathways [13]. Prolonged HG results in inefficient antioxidant defense, leading to oxidative damage precipitated by ROS and OxS, the primary causes of diabetic complications [14,15]. Elderly people are more vulnerable to OxS due to weakened endogenous antioxidant systems [16]. Aging increases nuclear factor-kappa B (NF-κB) activity via pathways such as mitogen-activated protein kinases (MAPKs) and IκB kinase (IKK), thereby fostering chronic inflammation and impairing the insulin signaling pathway [17]. Consequently, these processes contribute significantly to the initiation and progression of chronic diseases, including CMDs.

The antioxidant defense includes endogenous antioxidant enzymes such as superoxide dismutase (SOD), catalase (CAT), glutathione peroxidase (GPx), and low-molecular-weight antioxidants such vitamins C and E, polyphenols, and carotene [18]. The GPx family comprises eight distinct isoforms (GPx1-8) in humans, each with distinct substrate specificities and tissue distribution [19]. Among these isoforms, GPx3, although expressed primarily in the kidney and lung, is also released into the circulation where it can be quantified from serum samples. This approach allows for the non-invasive monitoring of GPx3 levels, providing a practical and ethical advantage in human studies. GPx3 is a selenoprotein that reduces hydrogen peroxide and lipid hydroperoxides, thereby playing a vital role in sustaining cellular homeostasis and preventing OxS. Moreover, GPx3 has garnered considerable attention due to its involvement in various pathological conditions, including diabetes, cardiovascular diseases, and cancer [20]. Several factors like hyperglycemia, selenium deficiency, polymorphisms in GPx3, aging, and lifestyle influence GPx3 levels, although the exact mechanisms remain unclear [21,22,23,24]. It is pertinent to acknowledge that the efficacy of antioxidant defenses declines progressively with advancing age [25].

MicroRNAs (miRNAs) are small, non-coding RNA molecules that are essential in the post-transcriptional regulation of gene expression [26]. MiRNAs influence up to 60% of human genes, interacting with target mRNA to modulate its degradation or inhibit translation [27,28]. Extensive recent studies have shown that miRNAs are intricately involved in diabetes-related processes, including endothelial dysfunction, insulin secretion, adipocyte differentiation, and pancreatic β-cell function [27,29]. Notably, these miRNAs are often dysregulated in key metabolic tissues such as the pancreas, white adipose tissue, and skeletal muscle, contributing to metabolic imbalances [27,30]. Furthermore, miRNAs are being investigated as potential biomarkers for T2DM due to their stability and consistent levels across individuals of the same ethnicity, which could revolutionize disease monitoring [31].

In addition to their role in metabolic regulation, miRNAs also interact with biochemical pathways influencing cellular health and disease progression. For example, hyperglycemia-induced alterations in miRNA expressions influence the level of molecules essential in antioxidant signaling such as Sirtuin-1 (SIRT1), forkhead box class O (FOXOs), Kelch-like ECH-associated protein 1/nuclear factor-erythroid 2 p45-related factor 2 (Keap1/Nrf2), SOD1/2, GPx-3, and CAT. This disruption can perpetuate a harmful cycle of elevated oxidative stress and further miRNA misregulation [27,32].

Based on an extensive literature review, including dual-luciferase reporter gene assays and predictive software analysis, we have identified miR-196a as a negative regulator of GPx3 [33]. This designation is reinforced by findings from Liu et al., which indicate that miR-196a downregulation enhances GPx3 expression, subsequently activating the c-Jun N-terminal kinase (JNK) pathway, pivotal in cellular processes like proliferation and apoptosis and critical for insulin signaling and TNF-α production [33,34]. Additionally, the interaction of miR-196a with the HOX8B 3′ untranslated region leads to its degradation, influencing glucose metabolism and adipose tissue regulation, underscoring its significant roles in both metabolic disorders [35,36,37].

In light of these findings and based on a comprehensive review of the literature, including results from dual-luciferase reporter gene assays and predictive software, we identified miR-196a as a negative regulator of GPx3. This selection is supported by its broad effects on gene regulation, which are pivotal in both metabolic disorders and cancer pathogenesis.

Given the rising incidence of CMDs in the aging population, our research aims to investigate the association between plasma GPx3 levels and miR-196a expression with metabolic disturbances in individuals aged 65 and older. By assessing the potential of these biomarkers, we seek to provide insights into their reliability in reflecting metabolic health and risk, paving the way for novel diagnostic and therapeutic strategies.

## 2. Results

### 2.1. Characteristics of the Patients

Table 1 shows a comparison of the studied groups (T2DM, prediabetes, control) in terms of anthropometric parameters: age, sex, height, blood pressure (BP), BMI, WHR, and all data gathered from BIA. The metabolic characteristics of the study group, including carbohydrate metabolism (fasting plasma glucose (FPG)), glycated hemoglobin (HbA1C), homeostasis model assessment for insulin resistance (HOMA-IR), triglycerides (TG)/high-density lipoprotein cholesterol (HDL), and lipid and renal parameters, are presented in Table 2.

The study groups did not differ significantly in terms of age, sex, height, systolic and diastolic blood pressure, and the following anthropometric parameters: BMI, WHR, visceral fat rating, muscle mass, fat mass, and total body water mass. As expected, the T2DM patients were characterized by significantly higher values of all three measured skinfold thicknesses (triceps, abdominal, thigh) and percentage of body fat compared to the control group. Additionally, as expected, in the T2DM group, markedly lower percentages of muscle tissue and total body water were found in relation to the control group. In addition, the T2DM patients had a thicker triceps skin fold and a higher percentage of body fat compared to the prediabetes patients.

In the carbohydrate metabolism analysis, both the prediabetes and T2DM groups exhibited significantly elevated FPG and HbA1c levels relative to the control group. The T2DM group, in particular, displayed markedly higher FPG levels compared to the prediabetes group, which also had significantly higher HbA1c levels compared to the controls. In terms of insulin resistance, a notable increase in HOMA-IR values was observed in the T2DM group relative to the control group. For lipid profile assessments, a significant reduction in HDL cholesterol levels was observed in the T2DM group compared to the controls, while other lipid parameters did not show significant variations. The observed decrease in low-density lipoprotein cholesterol (LDL) and total cholesterol (TC) levels in the T2DM group could be attributed to the administration of hydroxymethylglutaryl-CoA (HMG-CoA) reductase inhibitors known as statins. Regarding renal parameters, individuals in the T2DM group exhibited significantly elevated urea levels compared to the control group. Other parameters such as creatinine and the estimated glomerular filtration rate (eGFR) did not show significant differences across all the study groups.

Table 3 delineates the oral antihyperglycemic medications and lipid-modifying agents administered to the study population (n = 126). In the control group, 50% of the participants were prescribed statins. These agents, primarily utilized for the management of hypercholesterolemia, tend to reduce concentrations of TC, LDL, and TG while elevating the HDL level. In the prediabetes cohort, over half were on statin therapy. Notably, about 40% of individuals in the prediabetes group relied solely on dietary interventions, with a mere duo being prescribed metformin. Within the T2DM group, metformin emerged as the predominant therapeutic agent. Approximately one-third of these patients were on a combination of at least two agents (antidiabetic medications or HMG-CoA reductase inhibitors). Furthermore, an equivalent proportion consumed a regimen of at least three agents. Interestingly, individuals with T2DM were not administered medications known to influence the HOMA-IR index, such as sulfonylurea derivatives or glucagon-like peptide-1 receptor agonists (GLP-1 insulin analogs).

### 2.2. GPx3 Level

The level of GPx3 measured via ELISAs in the group with CMDs was statistically significantly lower than that in the control group (530 ± 353.5 vs. 413.3 ± 316.2, *p* < 0.05) (Figure 1a), which was the expected result. Then, the level of GPx3 in the T2DM and prediabetes groups was analyzed separately. Lower GPx3 values were observed in the patients with prediabetes and T2DM compared to the control group. However, statistical significance was demonstrated only between the control group and the group with T2DM (*p* < 0.05) (Figure 1b).

### 2.3. Relative Expression of miR-196a

The relative expression level of miR-196a in the control group and in the group with CMD is shown in Figure 2a, and that in the groups with prediabetes and T2DM is shown in Figure 2b. It was observed that the expression level of miR-196a did not differ statistically significantly among the study groups, although it was noted that in the groups with CMDs and T2DM, the level of expression was slightly increased in relation to the control group.

### 2.4. Correlation between GPx3 Level and miR-196a and Anthropometric and Metabolic Parameters

Spearman’s correlations between the level of GPx3 and miR-196a, anthropometric parameters, systolic blood pressure (SBP), and diastolic blood pressure (DBP) are presented in Appendix A. There were no statistically significant correlations between the level of GPx3 and miR-196a in the study groups. It is worth noting that although these correlations were not observed, one can see a statistically significant decrease in the level of GPx3 in the group with CMDs and T2DM and a trend toward an increase in miR-196a expression. In the control group, a negative moderate correlation was found between the level of GPx3 and DBP and the thickness of the triceps skinfold. On the other hand, in the group with CMDs, negative weak correlations were found between the level of GPx3 and the mass of adipose tissue, muscle, and total body water. The analysis of the prediabetes and T2DM groups showed only a negative moderate correlation between the level of GPx3 and SBP in the T2DM group.

Correlations between the level of GPx3 and metabolic and renal parameters are presented in Appendix A. Only moderate correlations between the level of GPx3 and renal parameters (creatinine, eGFR) were found in the control group. In the group with CMDs, a weak negative correlation was found between glucose and the GPx3 level. In addition, when analyzing the prediabetes and T2DM groups, this correlation was found to be stronger in the prediabetes group and absent in the T2DM group. This suggests that a significant decrease in the level of GPx3 in plasma occurs in the period preceding the development of full-blown T2DM, although such a suggestion was not confirmed by the level of GPx3 in the studied groups (Figure 1a,b). Moreover, there was a positive moderate correlation in the prediabetes group between the level of GPx3 and TG/HDL and a negative moderate correlation with HDL concentration.

Appendix A shows the correlations between the miR-196a expression level and the anthropometric parameters, SBP, and DBP. Among the anthropometric parameters, the only statistically significant weak positive correlation was observed between the percentage of body fat and the level of miR-196a expression in the group with CMDs.

Correlations between the level of miR-196a expression and metabolic and renal parameters are presented in Appendix A. In the control group, only negative moderate correlations were observed between the level of miR-196a expression and the concentration of total cholesterol and LDL cholesterol. No statistically significant correlations were found in the group with CMDs and T2DM. However, in the prediabetes group, a negative moderate correlation was observed between miR-196a and the fasting glucose level. The obtained results suggest that the downregulation of miR-196a may contribute to higher LDL and T-CH levels in the control group and higher FPG levels in patients with prediabetes.

### 2.5. Diagnostic Potential of Plasma GPx3 Level and miR-196a Expression in CMD Development Risk

ROC analysis with the AUC as a quality index was carried out to determine the potential of the GPx3 level as a diagnostic tool. Appendix A shows the results of the linear regression and correlation analysis between the GPx3 level and anthropometric and biochemical parameters in the whole study population. The plasma level of GPx3 was negatively correlated with SBP, HbA1c, glucose, creatinine, urea, eGFR, and mass of body fat assessed by BIA. The results of multivariate stepwise linear regression analyses showed that the values of SBP were the most strongly and significantly correlated with the plasma level of GPx3. This observation suggests a relationship between inefficient antioxidant defense and hypertension.

In our analysis, a prominent finding was the significant association of GPx3 with the outcome of CMDs, as evidenced in both univariate and multivariate logistic regression models, after excluding outliers due to the significant dispersion of GPx3 levels among the participants (Table 4). In the multivariate context, a unit increase (1 ng/mL) in the GPx3 level was associated with an approximate 0.002 reduction in the odds of CMDs occurring (0.2%, OR = 0.998; 95% CI: 0.996–0.999; *p* = 0.031). Thus, considering the GPx3 concentration ranges detected in the subjects, one can see that a 100 ng/mL increase in GPx3 decreases the odds ratio of CMDs by 0.2 (20%). Remarkably, this association retained its significance and directionality when the effect of GPx3 was evaluated in isolation, as revealed by the univariate analysis (OR = 0.998; 95% CI: 0.997–0.999; *p* = 0.042).

To further explore the potential of the GPx3 level as a diagnostic tool for CMD risk in the population aged 65 and older, ROC analysis was conducted (AUC = 0.6273; *p* = 0.0175) (Figure 3). The optimal cut-off point was determined using the Youden’s index method. The patients were divided into two subgroups according to the 419.501 ng/mL-cut-off point of the GPx3 level. The difference in the frequency of CMDs between the subgroups was tested using the chi-square test with the Yates correction (*p =* 0.0256). The risk of CMDs in the subgroup with a lower GPx3 level (<419.501 ng/mL) was found to be more than 5.2 times greater than that in the subgroup with a higher level of GPx3 (OR = 5.1754; CI: 2.31–11.58). To investigate the diagnostic potential of GPx3, we evaluated the CMD markers FPG and HbA1c, with the respective optimal cut-off points of 5.95 mmol/L (AUC 0.7436; *p* < 0.0001; OR = 12.209; 95% CI: 3.495; 42.656) and 5.70% (AUC 0.9465; *p* < 0.0001; OR = 729.2353; 95% CI: 41.0190; 12,964.3287). In turn, a comparison of the ROC curves revealed a significant difference between GPx3 and HbA1c (*p* < 0.0001) but a lack of a difference between GPx3 and FPG (*p* = 0.0877).

To check the potential of miR-196a expression as a diagnostic tool for CMD risk, ROC analysis was also performed. However, our results showed that miR-196a expression did not have a diagnostic value (AUC = 0.5017; *p* = 0.977).

## 3. Discussion

The steady increase in the prevalence of CMDs is strongly associated with overweight, obesity, and aging. The current diagnostic tools for CMDs (e.g., FPG, HbA1c, the glucose tolerance test, HOMA-IR, and anthropometric parameters) are useful only at the T2DM stage due to its silent nature. This delay in diagnosis leads to a higher risk of complications, particularly microvascular and macrovascular issues [1,38]. Therefore, many clinicians strongly emphasize the need for a prompt and early diagnosis of CMDs. Given the rising incidence of CMDs among the aging population, our study was designed to explore the relationship between the GPx3 plasma levels, a key component of the antioxidant defense system, and the expression of miR-196a, which negatively regulates mRNA GPx3. We made every effort to ensure that the study groups did not differ significantly from each other in terms of age, sex, and most of the anthropometric parameters (body weight, BMI, and WHR). The only variable anticipated to influence the levels of GPx3 and the expression of miR-196a was increased glycemia.

The results of our study suggest that reduced GPx3 levels may serve as a prognostic marker for CMDs in people over 65 years of age. The ROC analysis demonstrated that the risk of CMDs in the group with a GPx3 level of less than 419.501 ng/mL was 5.2 times higher than that in the group with a GPx3 level higher than 419.501 ng/mL (CI: 2.31–11.58, respectively; OR = 5.175). However, despite initial enthusiasm, this observation was primarily attributable to the significantly lower GPx3 levels found in the T2DM group. In contrast, the reduction in GPx3 in the prediabetes group did not reach statistical significance when compared to the control group. Therefore, the lack of a clear decline in the GPx3 levels in people with prediabetes suggests that GPx3 may not be a reliable early indicator of CMDs. Although it cannot be ruled out that a larger sample size might have achieved statistical significance, the current data do not support the use of GPx3 as an early predictor of CMDs. However, GPx3 may still hold potential as an indicator of progression to diabetes.

The decreased GPx3 level in the patients with T2DM aligns with the previous research findings of Ling et al., Chung et al. Kaliaperumal et al., and Sedigihi et al. [39,40,41,42]. In contrast to our findings, Langhardt et al. did not demonstrate a clear difference in GPx3 levels between the T2DM and normoglycemic groups, but they noted that GPx3 levels might be influenced by weight changes and fasting plasma insulin levels after bariatric surgery [43]. Hauffe et al. identified diminished GPx3 levels in T2DM patients, which correlated with reduced insulin receptor expression and insulin sensitivity in adipose tissue in multiple mouse models [21]. Flehmig et al. revealed that GPx3 is a member of a cluster of adipokines, which is closely related to insulin sensitivity, hyperglycemia, lipid metabolism, and inflammation in humans [44]. Additionally, the *GPX3* promoter features binding sites for regulatory elements like Sp-1 (specificity protein 1) and HIF-1 (hypoxia-inducible factor-1) as well as a metal response element (MRE) and an antioxidant response element (ARE) [20,45]. Therefore, further research in this field is required.

In prediabetes probands, we observed a trend toward a GPx3 decrease. Taking into account the natural development of T2DM, which is preceded by many years of prediabetes, it seems that the level of GPx3 may slightly decrease. We hypothesize that the relatively low number of participants in the prediabetes group, coupled with a higher proportion of individuals using statins and adhering to a diabetic diet compared to the control group, could have potentially influenced the observed outcomes. Furthermore, a meta-analysis by Zinellu A et al. suggests that statins may increase GPx3 levels [46].

In our research, the obtained results indicated that expression of miR-196a was slightly, but not significantly, elevated in the CMD and T2DM groups in comparison to the control subjects. Perhaps the small number of subjects in the studied groups limited the observed effect. Nevertheless, despite that, in line with our observations, previous research has confirmed that miR-196a expression is upregulated in T2DM patients compared to normoglycemic probands [47,48].

Given the results by other groups confirming, via luciferase assays, that miR-196 targets GPx3 mRNA [33], a correlation analysis was conducted between these markers and various clinical parameters to enhance the robustness of this study. Although no significant correlation was found between GPx3 and miR-196a levels, a notable decrease in GPx3 levels in the CMD and T2DM groups and a trend toward increased miR-196a expression were observed [33]. It should be recognized that mechanisms other than miR-196a might affect the GPx3 level. Firstly, histone acetylation or methylation, particularly under the influence of hypoxia, a condition prevalent in excess adipose tissue, increases genome-wide bivalent epigenetic marking, specifically enhancing H3K27me3 levels, which can impact gene regulation [49]. Secondly, GPx3 protein levels may be influenced by various post-translational modifications, including oxidative modifications of cysteine, phosphorylation, acetylation, ubiquitination, glycosylation, protein fragmentation, and protein misfolding [50]. Notably, glycosylation is the most common modification observed in T2DM individuals [51].

In our study, we observed limited correlations between GPx3 levels and anthropometric and metabolic parameters across the different groups. In the prediabetes and T2DM groups, the only significant finding was a moderate negative correlation between GPx3 levels and SBP, particularly in the T2DM subgroup. We noted moderate correlations between GPx3 levels and renal parameters (creatinine and eGFR) in the control group, consistent with previous findings [52,53,54]. GPx3 insufficiency could exacerbate OxS in the vascular endothelium, instigating inflammation mediated by factors like NF-κB and accelerating glycation processes, leading to podocyte injury and compromising glomerular filtration [52,55]. Moreover, OxS might also activate fibrotic pathways (e.g., TGF-β/Smad), contributing to renal fibrosis and declining eGFR [56]. Additionally, an inverse relationship was found between FPG and GPx3 in the CMD and prediabetes groups. No correlations were observed between HbA1c or HOMA-IR and GPx3 levels, aligning with prior studies [39,42]. Surprisingly, a positive correlation emerged between GPx3 levels and the TG/HDL ratio in the prediabetes group. The significance of this correlation remains uncertain and warrants further investigation.

Interestingly, obesity, a key precursor to IR, is linked to adipose tissue dysfunction, chronic inflammation, and the imbalanced secretion of adipokines such as leptin and adiponectin, which diminish insulin sensitivity [34,57]. Researchers continue to search for affordable and simple anthropometric and metabolic parameters that accurately reflect the metabolic condition of individuals. Our study also aimed to identify the most effective anthropometric parameter for reflecting the metabolic state of elderly patients. It is notable that simple measurements like BMI, WHR, and skinfold thickness correlate strongly with metabolic disorders, with bioimpedance analysis (BIA) providing more detailed assessments [58,59,60]. In our study, a negative correlation was observed between triceps skinfold thickness and GPx3 levels in the control group, though no such associations were found for other skinfold sites or within the CMD, prediabetes, and T2DM groups. Despite several significant correlations between GPx3 levels and parameters indicative of overweight and obesity, traditional metabolic risk factors like WHR, BMI, and body weight showed no significant correlations. These results were anticipated due to the homogeneity of the study groups concerning WHR, BMI, and body weight. This finding is consistent with Ling et al., who also reported no association between BMI and GPx3 levels [39]. In the CMD group, there were only weak negative correlations between GPx3 levels and fat mass, muscle mass, and total body water.

To the best of our knowledge, this is the first study to examine miR-196a expression in pre-diabetes patients. Our findings reveal isolated correlations between miR-196a expression and both metabolic and anthropometric parameters, suggesting that miR-196a may not effectively reflect the overall metabolic status. In the control group, miR-196 expression was moderately negatively correlated with TC and LDL cholesterol. Conversely, in the pre-diabetes group, miR-196 expression was positively correlated with FPG concentration. Previous studies have also highlighted the role of miR-196a in metabolic regulation; Hilton et al. concluded that miRNA-196a is likely involved in regulating body fat mass [61], while Mori et al. demonstrated that miR-196a targets the CCAAT-enhancer-binding protein (CEBPβ), which subsequently regulates the brown adipogenesis signal HOXC8 [62].

Our study has limitations, including the lack of cellular GPx activity measurements in our patients. Previous analyses have shown a positive correlation between oxidative stress and cellular GPx activity in individuals with T2DM [63]. Moreover, the size of the research group, especially the control group, was limited due to challenges in identifying hospitalized patients who met the stringent inclusion criteria for the control group. The analysis also did not adjust for gender because of the number of participants in each group. Additionally, due to financial constraints, we did not measure the expression of the GPX3 gene itself, which could have enhanced the methodological rigor of the study.

The strengths of this study are as follows: (1) human samples were collected from the elderly population, a demographic seldom represented in miRNA studies; and (2) substantial data were gathered on anthropometric and metabolic parameters, providing a valuable resource for future research and meta-analyses. However, additional clinical studies on a larger number of samples are required to assess whether decreased GPx3 is involved in the development of T2DM.

## 4. Materials and Methods

### 4.1. Study Design

Among 131 preliminarily qualified participants, 126 were incorporated into the study. These patients were admitted between January 2019 and February 2022 for a range of internal medicine conditions at the Department of Internal Medicine, Diabetology, and Clinical Pharmacology of the Medical University of Lodz. This study received the endorsement of the Bioethics Committee at the Medical University of Lodz (document reference: RNN/193/18/KE; approval date: 18 May 2018) and was executed in strict adherence to the principles of Good Clinical Practice and the Declaration of Helsinki. Prior to enrollment, each participant provided written consent to participate in the study.

The criteria for participant inclusion in the CMD group were as follows: age ≥ 65 years, diagnosed with T2DM or taking diabetes medications, diagnosis of T2DM or prediabetes as per the criteria delineated by the American Diabetes Association (ADA 2019), or a glycated hemoglobin (HbA1c) level of ≥5.7 or fasting plasma glucose (FPG) level of >5.5 mmol/L. The ADA has defined four diagnostic modalities for diabetes, and these modalities are concurrently utilized for prediabetes [64]. The HbA1c test, preferred for diagnosing T2DM and prediabetes, reflects average blood glucose over three months, requires no fasting, and is less influenced by daily glucose fluctuations compared to FPG [65].

Participants aged over 65 years exhibiting HbA1c < 5.7% and an FPG measurement of ≤5.5 mmol/L were designated for inclusion within the control group. The study’s exclusion criteria encompassed patients aged below 65 years, those diagnosed with diabetes other than T2DM, and individuals with a presence or historical diagnosis of malignancies. Additionally, those with prior exposure to radiotherapy or chemotherapy, acute medical conditions characterized by elevated CRP or pronounced leukocytosis, and clinical exigencies such as acute coronary syndrome, acute abdominal complications, and exacerbations of conditions like COPD or asthma were omitted. Active smokers, those with a familial history indicating genetic predispositions or documented genetic disorders, those with decompensated thyroid pathologies like hyperthyroidism and hypothyroidism, and individuals with profound hepatic ailments or a recent history of blood transfusion were also excluded. Subsequent to their enrollment, each patient’s medical background was meticulously reviewed, complemented by a comprehensive physical examination. Among the initially selected participants, five were rendered ineligible for the study: two owing to the detection of malignancies, and three due to sepsis.

Qualified participants were assigned to a group with CMDs (n = 88) and a control group (without CMDs, n = 38). The CMD group included patients with prediabetes (n = 37) and T2DM (n = 51). Prediabetes was characterized by FPG levels between 5.6 and 6.9 mmol/L or HbA1c values of 5.7–6.4%. The T2DM group consisted of individuals with a prior diagnosis of diabetes or currently under hypoglycemic medication treatment or exhibiting FPG levels of >7 mmol/L or HbA1c levels of ≥6.5%. The criteria for allocating patients to the prediabetes and T2DM subgroups are presented in Figure 4.

Blood pressure (BP, including systolic blood pressure—SBP—and diastolic blood pressure—DBP) and anthropometric parameters (body weight, height, waist circumference (WC), and hip circumference (HC)) were measured and used to calculate the body mass index (BMI) and waist–hip ratio (WHR). Next, blood samples were taken to determine FPG, insulin, HbA1c, total cholesterol (T-CH), LDL cholesterol (LDL-CH), HDL cholesterol (HDL-CH), triglycerides (TG), uric acid, and creatinine concentrations. The estimated glomerular filtration rate (eGFR) was calculated based on the Modification of Diet in Renal Disease (MDRD) equations. Serum FPG, HbA1c, TC, HDL, LDL, TG, and insulin levels were assessed using standard analytical methods. HbA1c levels were determined by means of high-performance liquid chromatography (HPLC method), and FPG levels were measured photometrically under ultraviolet light using hexokinase-catalyzed enzymatic reactions. The enzyme colorimetric assay was quantified for TC, HDL, and TG using Beckman Coulter AU analyzers (Beckman Coulter, Brea, CA, USA). LDL was calculated using the Friedewald formula. The insulin concentration was determined via electrochemiluminescence testing. Based on the obtained results, the homeostatic model assessment of insulin resistance (HOMA-IR) and TG/HDL ratio were calculated [66]. The formula for HOMA-IR is as follows: HOMA-IR = (FPG (mmol/L) × FPI (pmol/L))/22.5.

Fasting venous blood plasma was separated for further analysis of microRNA expression and measurement of GPx3. A whole-blood sample was centrifuged at 3000 x g for 10 min at 4 °C in a centrifuge. A 200 µL aliquot of plasma was carefully combined with a QIAzol^®^ lysis reagent (Qiagen, Germantown, MD, USA), and the mixture was promptly stored at −80 °C in preparation for subsequent isolation procedures. This process was conducted according to the manufacturer’s guidelines specified in the miRNeasy Serum/Plasma Kit (Qiagen, USA). The remaining part of the plasma, which was used to measure the level of GPx3, was divided into several 100 µL tubes and stored at −80 °C until the ELISA test was performed.

Between 8:00 and 10:00 am. after an overnight fast, the following anthropometric measurements were gathered: weight, height, WC, and HC of the participants. Based on the obtained data, the BMI and WHR were calculated. In addition, using a skinfold caliper (BATY CE 0120, Baty International, Burgess Hill, UK), the skinfold thickness was measured according to the manufacturer’s instructions in three places: 1—above the triceps muscle of the non-dominant arm, 2—in the umbilical region (2 cm below the navel in the sternum line), 3—above the muscle quadriceps in all participants using BATY^®^ skinfold calipers. The measurement precision was 0.20 mm, the accuracy was 99.00, and the measuring range was from 0 to 80 mm. Moreover, using bioelectrical impedance analysis (BIA method) with an electrical scale (TANITA MC-780MA, TANITA Corporation, Sportlife Tokyo, Japan; technical error of measurement (TEM = 0.719) with a range of 0–280 kg and accurate to within 0.1 kg) with a built-in eight-electrode system that measured impedance at multiple frequencies in the range from 5 kHz to 0.25 MHz, the body composition of the probands was assessed. The measurement was taken during an examination in underwear without clothes and socks, with the weight evenly distributed on the feet. The obtained result was related to the standards adopted for age, height, and sex, taking into account the percentage and mass of the body composition. Measurements were made with an accuracy of 0.1 kg when determining body weight, muscle mass, fat, and total body water, and with an accuracy of 0.1% when determining the percentage of the body composition. Segmental analysis included body fat mass (BF), muscle mass–free fat body mass (FFM), and calculated visceral fat rating (calculated by the manufacturer’s software; scores 1–12 were considered healthy; scores of 13–59 indicated excess visceral fat).

### 4.2. Determination of GPx3—ELISA

Plasma GPx3 levels were assessed using an ELISA kit from Wuhan, China. Plasma samples were subjected to a 70-fold dilution with 0.01 mol/L PBS buffer. The ELISA was conducted according to the manufacturer’s instructions.

### 4.3. Quantitative Real-Time PCR Assay (qRT-PCR)

A total of 2 μL of isolated total RNA was subjected to a cDNA synthesis reaction carried out using a miRCURY LNA RT Kit (Qiagen, USA) according to the manufacturer’s instructions. cDNA synthesis was performed for 60 min in a GeneAmp PCR System 9700 thermal cycler (Applied Biosystems, Foster City, CA, USA) at 40 °C for 5 min followed by at 95 °C to inactivate the reverse transcriptase. The reaction mixture was cooled to 4 °C and immediately used for the qPCR reaction or stored at −20 °C.

Expression profiling was performed with a miRCURY LNA miRNA Probe PCR Assay (Qiagen, USA) in a 7900HT Fast Real-Time PCR System (Applied Biosystems, Foster City, CA, USA) under the following conditions: 2 min at 95 °C, 40 cycles of 5 s at 95 °C, and 30 s at 56 °C. Data were analyzed using the SDS 2.4 software and Data Assist software v3.01 (Applied Biosystems, USA). Data were normalized using the 2^−ΔCt^ method with the arithmetic average of Ct values for 2 reference genes: miR-16-5p and miR-103a-3p. The assay IDs (Applied Biosystems, Foster City, CA, USA) for the analyzed molecules were as follows: has-miR-196a-5p (sequence catalog number: ZP00000381, sequence: UAGGUAGUUUCAUGUUGUUGGG), has-miR-103a-3p (sequence catalog number: ZP00000028, sequence: AGCUUCUUUACAGUGCUGCCUUG), and has-miR-16-5p (sequence catalog number: ZP00000315, sequence: UAGCAGCACGUAAAUAUUGGCG). The expression levels of the normalized miRNAs are shown by their median and upper and lower quartiles (25–75th percentile), as their distribution was not Gaussian. Has-miR-103a-3p and has-miR-16-5p were used as the endogenous control against which the expression of the studied miR-196a was assessed [67]. The control miRNAs had a stable level of expression in the blood [68].

### 4.4. Statistical Analysis

The anthropometric and biochemical characteristics as well as the GPx3 level and relative expression of miR-196a of the groups were expressed as medians with lower and upper quartiles. As the distribution of variables did not agree with normality, as determined via the Shapiro–Wilk test, the differences between the 2 groups were assessed using the Mann–Whitney U test. The relationship between the GPx3 level, expression of miR-196a, and anthropometric and biochemical parameters was evaluated using a Spearman (non-parametric) correlation coefficient. The most significant GPx3- and miR-196a-related variables were identified using the stepwise forward multiple regression method. Logistic regression analysis was conducted to assess the relationship between the risk of CMDs and the GPx3 level, as well as the relative expression of miR-196a. The non-parametric variables included in the linear regression analysis were logarithmically transformed. Receiver operating characteristic (ROC) analysis with an area under the curve (AUC) as a quality index was carried out to determine the potential of the GPx3 level as a diagnostic tool, as well as the relative expression of miR-196a. The optimal cut-off point for the GPx3 level was determined using Youden’s index. The patients were divided into two subgroups according to the cut-off point for the GPx3 level. The difference in the frequency of CMDs between the subgroups was tested using the chi-square test with the Yates correction. All analyses were performed using the GraphPad Prism Software 8.0 (San Diego, CA, USA), STATISTICA Software 13.3 (TIBCO, Palo Alto, CA, USA), and MedCalc Statistical Software 22 (MedCalc Software Ltd., Ostend, Belgium). A level of *p* < 0.05 was considered statistically significant.

## 5. Conclusions

Our study highlights the association between GPx3 plasma levels and miR-196a expression in elderly patients with CMDs. The obtained results demonstrate that although decreased GPx3 levels could indicate CMD in T2DM patients, they fail to consistently signal the earlier stages, such as prediabetes. Our results show that microRNA-196a is not a good diagnostic marker for glucose metabolism disorders. Further studies with larger, diverse cohorts are needed to explore whether a decreased plasma level of GPx3 in CMD patients may be a result of the upregulation of miR-196a.

## Figures and Tables

**Figure 1 ijms-25-05409-f001:**
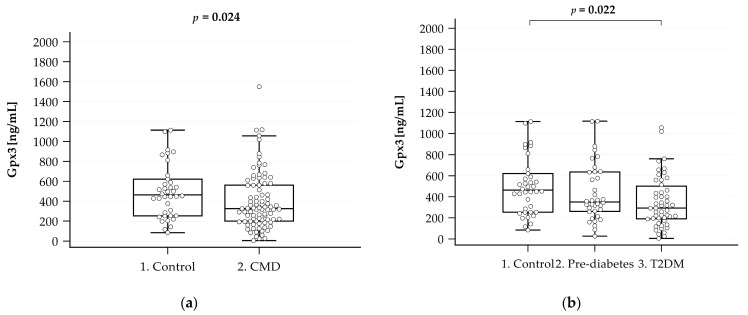
Plasma levels of GPx3 in individuals aged 65 and older from the control group (n = 38) and CMD group (n = 88) (**a**) and in the control (n = 38), prediabetes (n = 37), and T2DM (n = 51) individuals aged 65 and older (**b**). GPx3 levels were measured via ELISA. Data are expressed as medians with lower and upper quartiles and minimum and maximum values in the form of box-and-whisker plots overlaid with dot-blot charts.

**Figure 2 ijms-25-05409-f002:**
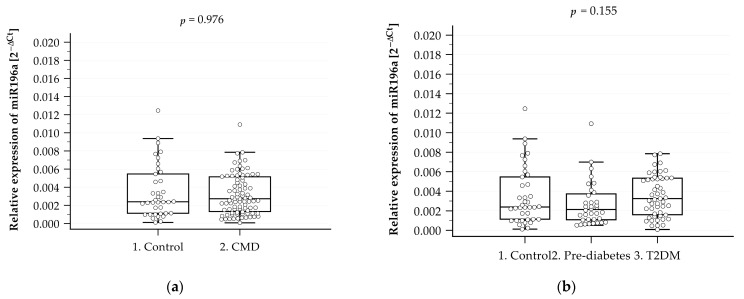
Level of miR-196a in plasma in the control group (n = 38) and CMD patients (n = 88) aged 65 and older (**a**) and in the control (n = 38), prediabetes (n = 37), and T2DM (n = 51) patients aged 65 and older (**b**). miRNA expression was assessed by RT-qPCR. Data are expressed as medians with lower and upper quartiles and minimum and maximum values in the form of box-and-whisker plots overlaid with dot-blot charts.

**Figure 3 ijms-25-05409-f003:**
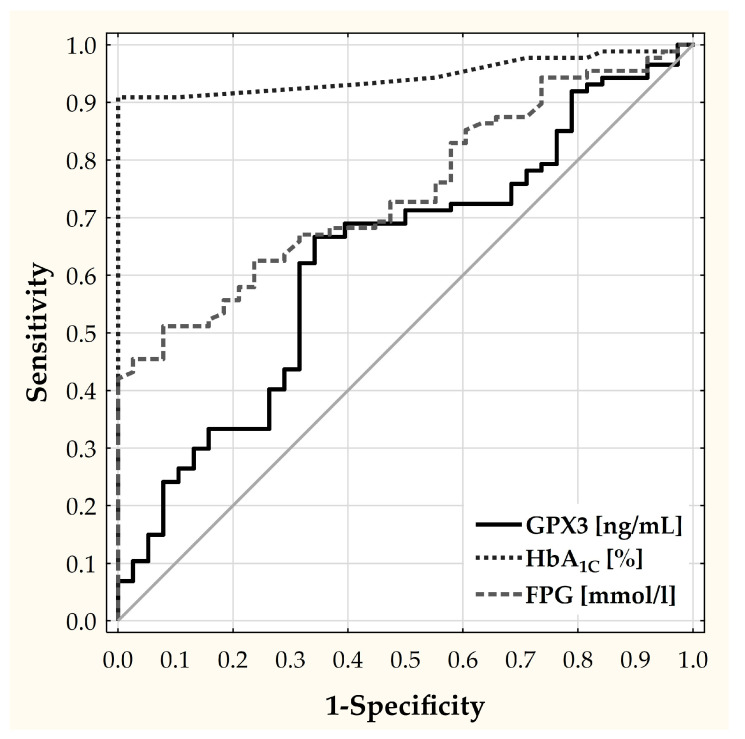
Receiver operating characteristic (ROC) curve of GPx3, HbA1c, and FPG in distinguishing CMD risk in subjects aged 65 and older.

**Figure 4 ijms-25-05409-f004:**
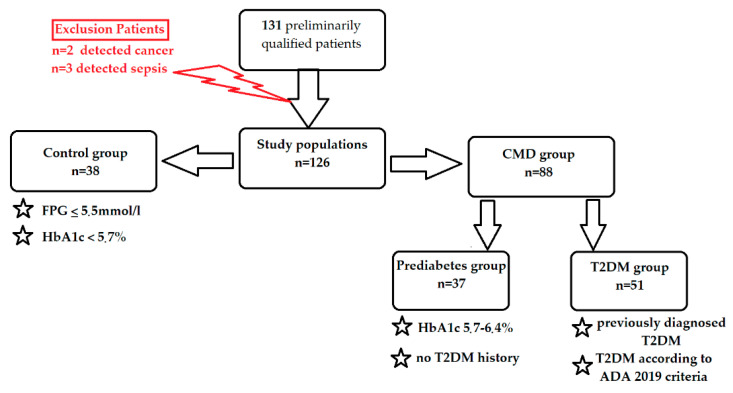
Overall flowchart of study design.

**Table 1 ijms-25-05409-t001:** Anthropometric parameters and blood pressure values of patients from the control group (n = 38), prediabetes group (n = 37), and T2DM group (n = 51) aged 65 and older.

Parameters	Control (n = 38)	Prediabetes (n = 37)	T2DM (n = 51)
F/M	18/20	22/15	34/17
Age [years]	73.00 (67.75; 85.00)	76.00 (69.5; 85.5)	75.00 (70.00; 82.00)
SBP (mm Hg)	130.00 (120.00; 140.00)	130.00 (123.50; 139.50)	130.00 (125.00; 142.00)
DBP (mm Hg)	79.00 (70.00; 80.00)	80.00 (70.00; 83.50)	80.00 (70.00; 80.00)
Body mass [kg]	77.65 (62.75; 85.00)	73.80 (61.45; 80.00)	71.70 (61.00; 87.60)
Height [m]	1.68 (1.56; 1.76)	1.63 (1.53; 1.72)	1.62 (1.52; 1.70)
BMI [kg/m^2^]	26.26 (23.68; 31.14)	26.50 (24.40; 29.25)	27.25 (24.93; 30.93)
WC [cm]	102.50 (92.50; 110.50)	98.00 (94.00; 105.80)	103.00 (94.00; 112.00)
HC [cm]	98.00 (90.00; 108.00)	98.00 (92.50; 103.00)	97.00 (94.00; 108.00)
WHR	1.04 (0.95; 1.07)	1.00 (0.97; 1.05)	1.03 (0.99; 1.06)
ST triceps (mm)	**15.60 (10.80; 20.20)**	**17.40 (11.10; 24.50)**	**21.80 (17.80; 27.60) ^bb,c^**
ST abdominal (mm)	**26.00 (18.70; 33.55)**	31.00 (19.60; 37.90)	**34.00 (26.80; 41.40) ^b^**
ST thigh (mm)	**18.40 (14.35; 25.20)**	23.40 (11.80; 36.20)	**31.20 (17.80; 38.80) ^b^**
Visceral fat rating	11.00 (8.75; 12.25)	11.50 (9.00; 14.00)	12.50 (9.75; 14.00)
BIA–BF [%]	**25.00 (20.70; 31.10)**	**25.00 (20.70; 32.50) ^c^**	**30.60 (24.10; 37.80) ^b,^** ^c^
BIA–BF [kg]	18.60 (12.90; 23.93)	18.75 (14.43; 23.40)	20.70 (16.20; 29.30)
BIA–FFM [%]	**71.00 (64.65; 74.65**)	70.20 (62.88; 75.28)	**65.90 (59.50; 72.00) ^b^**
BIA–FFM [kg]	54.00 (45.13; 60.30)	49.50 (40.70; 57.81)	47.70 (38.48; 61.08)
BIA–TBW [%]	**52.30 (48.60; 56.70)**	52.20 (46.90; 55.70)	**48.70 (44.50; 51.80) ^b^**
BIA–TBW [kg]	40.10 (32.10; 46.78)	37.15 (29.50; 43.20)	35.90 (28.30; 43.88)

List of abbreviations: F—females, M—males, T2DM—type 2 diabetic subjects, BMI—body mass index, WHR—waist–hip ratio, WC—waist circumference, HC—hip circumference, ST—skinfold thickness, BIA—bioelectrical impedance analysis, BF—body fat, FFM—free fat mass–muscle mass, TBW—total body water. Statistically significant results are shown in bold. ^bb^ *p* < 0.01; ^b^ *p* < 0.05 control vs. T2DM; ^c^ *p* < 0.05 prediabetes vs. T2DM.

**Table 2 ijms-25-05409-t002:** Metabolic parameters of patients in the control group (n = 38), prediabetes group (n = 37), and T2DM group (n = 51) aged 65 and older.

Parameters	Control (n = 38)	Prediabetes (n = 37)	T2DM (n = 51)
HbA1c [%]	**5.50 (5.20; 5.60)**	**6.00 (5.80; 6.20)** ^aaa^	**6.50 (5.80; 7.60**) ^bbb^
FPG [mmol/L]	**5.16 (4.67; 5.37)**	**5.37 (4.94; 5.90)**	**6.71 (5.78; 9.55)** ^bbb,ccc^
HOMA-IR	**1.88 (1.15; 2.83)**	2.41 (1.38; 3.80)	**2.68 (1.86; 4.86**) ^b^
TG/HDL ratio	1.09 (0.67; 1.51)	0.99 (0.62; 1.38)	1.16 (0.75; 1.75)
Creatinine [μmol/L]	87.52 (71.90; 110.60)	89.00 (74.15; 109.80)	95.60 (69.55; 113.50)
Urea [mmol/L]	**5.85 (4.50; 7.64)**	7.08 (5.31; 9.38)	**7.67 (6.02; 9.54)** ^b^
eGFR [mL/min/1.73 m^2^]	70.69 (44.40; 90.40)	61.60 (44.00; 85.60)	55.20 (44.30; 83.40)
LDL [mmol/L]	2.53 (1.79; 3.42)	2.15 (1.75; 2.85)	2.13 (1.48; 3.14)
HDL [mmol/L]	**1.29 (1.01; 1.49)**	1.19 (0.95; 1.55)	**1.12 (0.87; 1.29) ^b^**
TG [mmol/L]	1.32 (0.82; 1.72)	1.08 (0.86; 1.54)	1.25 (0.91; 1.70)
TC [mmol/L]	4.47 (3.74; 5.11)	4.04 (3.32; 4.70)	3.64 (2.95; 5.21)

List of abbreviations: HbA1c—glycated hemoglobin FPG—fasting plasma glucose, HOMA-IR—homeostasis model assessment for insulin resistance, eGFR—estimated glomerular filtration rate, TC—total cholesterol, LDL-C—low-density lipoprotein cholesterol, HDL-C—high-density lipoprotein cholesterol, TG—triglycerides. Statistically significant results are shown in bold. ^aaa^ *p* < 0.001 control vs. prediabetes; ^bbb^ *p* < 0.001; ^b^ *p* < 0.05 control vs. T2DM; ^ccc^ *p* < 0.001 prediabetes vs. T2DM.

**Table 3 ijms-25-05409-t003:** Oral antihyperglycemic and lipid-modifying drugs used by all study participants (n = 126).

Name of the Agent	Control (n = 38)		Prediabetes (n = 37)		T2DM (n = 51)	
	*n*	*%*	*n*	*%*	*n*	*%*
Metformin (1–3 g/day)	0	0.00	2	5.41	38	74.51
TZD–pioglitazone (15 mg/day)	0	0.00	0	0.00	8	15.69
iSGLT2–empagliflozin (10 mg/day)	0	0.00	0	0.00	11	21.57
αGI–acarbose (150–300 mg/day)	0	0.00	0	0.00	9	17.65
Atorvastatin (10–40 mg/day)	7	18.42	17	45.95	33	64.71
Rosuvastatin (10–40 mg/day)	12	31.58	5	13.51	12	23.53
Fenofibrate (267 mg/day)	0	0.00	0	0.00	4	7.84
Diabetes diet treatment only	0	0.00	15	40.54	1	1.96
1 agent	0	0.00	20	54.05	10	19.61
2 agents	0	0.00	2	5.41	19	37.25
3 agents	0	0.00	0	0.00	17	33.33
4 agents	0	0.00	0	0.00	4	7.84

List of abbreviations: Metformin—in various doses from 1 g to 3 g daily; iSGLT2—sodium-glucose cotransporter-2 inhibitor; TZD—thiazolidinedione; αGI—α-glucosidase inhibitor; 1 agent—one drug from the table; 2 agents—two drugs from the table; 3 agents—three drugs from the table; 4 agents—four drugs from the table.

**Table 4 ijms-25-05409-t004:** Multivariate and univariate logistic regression analyses for carbohydrate metabolism disorders prognosis in subjects aged 65 and older.

Variable	Multivariate			Univariate		
	Wald’s P	OR	95% CI	Wald’s P	OR	95% CI
GPx3	0.031	0.998	0.996–0.999	0.042	0.998	0.997–0.999
SBP	0.795	0.996	0.969–1.024			
Creatinine	0.764	0.997	0.980–1.015			
Urea	0.578	1.048	0.887–1.239			
BIA-BF [%]	0.217	1.036	0.980–1.095			
Sex (Male)	0.466	0.838	0.521–1.348			

List of abbreviations: GPx3—glutathione peroxidase 3; SBP—systolic blood pressure; BIA-BF—bioimpedance analysis–body fat; OR—odds ratio; CI—confidence interval; SE—standard error.

## Data Availability

The data presented in this study are available on request from the corresponding author.

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
