# Peer review of "Association of Glutathione Peroxidase 3 (GPx3) and miR-196a with Carbohydrate Metabolism Disorders in the Elderly"

_ijms, 2024, doi:10.3390/ijms25105409_

Round 1

Reviewer 1 Report

Comments and Suggestions for Authors

Association of glutathione peroxidase 3 (GPx3) and miR-196a with carbohydrate metabolism disorders related anthropometric and metabolic parameters

The present study aimed to demonstrate the early diagnosis of CMD by analyzing GPx3 and miR 196a as plasma biomarkers. The approach and the overall design of the study are weak. The weakness that dampened the enthusiasm was a general lack of clarity in the overall presentation of the ideas and a lack of logical flow of the sentence structures and ideas/results, which falls behind in overall good scientific writing.

GPx3 levels showed significant change in the T2DM group only, as compared to the control group, not in the pre-diabetic group. Then how can GPx3 be used as an early marker of CMD diagnosis?

TG/HDL ratio was non-significant between control and T2DM. Then what will be the inference for the positive correlation between GPx3 Vs TG/HDL in prediabetic and control groups?

Overall, the study failed to demonstrate a significant correlation that logically links GPx3 and miR196a with metabolic parameters of CMD in prediabetic and T2DM populations for their use as early diagnostic markers. A better population size with a significant statistical correlation is required to conclude the aim.  

Comments on the Quality of English Language

Minor editing of the English language is required.

Author Response

Response to Reviewer 1:

Comment: The present study aimed to demonstrate the early diagnosis of CMD by analyzing GPx3 and miR 196a as plasma biomarkers. The approach and the overall design of the study are weak. The weakness that dampened the enthusiasm was a general lack of clarity in the overall presentation of the ideas and a lack of logical flow of the sentence structures and ideas/results, which falls behind in overall good scientific writing.

Response: Thank you for your constructive criticism regarding the clarity and structure of our manuscript. We acknowledge the importance of a clear and logically structured presentation in scientific writing. In response to your feedback, we have revised the manuscript to enhance its consistency and logical flow. The introduction and discussions have been refined and succinctly streamlined to ensure a more coherent progression of ideas. Moreover, based on your suggestion, we revisited our conclusions concerning the biomarkers GPx3 and miR 196a. Indeed, our revised analysis confirms that while a significant effect was observed in the CMD group and the T2DM subgroup, GPx3 and miR 196a did not exhibit the same significance as early markers in the prediabetes group. We have updated our conclusions and discussion sections to reflect these findings more accurately.

To provide more context, our revised text now states: " Our study highlights the association between GPx3 plasma levels and miR-196a expression in elderly patients with CMDs. The obtained results demonstrate that although decreased GPx3 levels could indicate CMD in T2DM patients, they fail to consistently signal the earlier stages, such as prediabetes. Our results show that microRNA-196a is not a good diagnostic marker for glucose metabolism disorders. Further studies, with larger, diverse cohorts, are needed to explore whether the decreased plasma level of GPx3 in CMD patients may be a result of the upregulation of miR-196a (lines 564-570)

Comment: GPx3 levels showed significant change in the T2DM group only, as compared to the control group, not in the pre-diabetic group. Then how can GPx3 be used as an early marker of CMD diagnosis?

Response:

Thank you for this  comment. Our conclusion was too far reaching. We fully agree with your comment. Although, we found significant decrease of GPX3 in CMD group (p< 0.024), which contains prediabetes and T2DM patients, the analysis in subgroups showed only significant decrease of GPX3 in T2DM, not prediabetes subgroup. Therefore, the results obtained did not clearly indicate that GPx3 may be an early biomarker of CMD. We reported  insignificant diminish in GPx3 level in prediabetes group, but it could be a result of not only early hyperglycemia (metabolic dysfunction) but also taking medications such as statins  or a diabetic diet that also affect GPx3 level. We have introduced relevant changes throughout the paper, highlighted in red (Abstract: lines 30-32; 35-38; Introduction: lines 52-57, 67-72, 77-80, 86-88, 91-100, 104-124; Discussion: lines 317-322, 335-340, 352-362, 374-375, 378-381, 389-393, 413-416; and Conclusion). Additionally, we have shortened parts of the introduction and discussion by removing unnecessary information.  We have changed our tone in the discussion section where we claim that: “The results of our study suggest that reduced GPx3 levels may serve as a prognostic marker for CMDs in people over 65 years of age. The ROC analysis demonstrated that the risk of CMDs in the group with a GPx3 level of less than 419.501 ng/ml is 5.2 times higher than that in the group with a GPx3 level higher than 419.501 ng/ml (CI:2.31 - 11.58, respectively; OR = 5.175). However, despite initial enthusiasm, this observation is primarily attributable to the significantly lower GPx3 levels found in the T2DM group. In contrast, the reduction in GPx3 in the prediabetes group did not reach statistical significance when compared to the control group. Therefore, the lack of a clear decline in the GPx3 levels in people with prediabetes suggests that GPx3 may not be a reliable early indicator of the CMDs. Although it cannot be ruled out that a larger sample size might have achieved statistical significance, the current data do not support the use of GPx3 as an early predictor of CMD. However, GPx3 may still hold potential as an indicator of progression to diabetes.” ( Lines 310-322)

Comment: TG/HDL ratio was non-significant between control and T2DM. Then what will be the inference for the positive correlation between GPx3 Vs TG/HDL in prediabetic and control groups?

Response: Thank you for your insightful comment concerning the observed patterns in our study between the TG/HDL ratio and GPx3 levels across different groups. The absence of a significant difference in the TG/HDL ratio between the control group and those with diabetes likely stems from the treatment regimen followed by all T2DM patients in the study, which predominantly includes metformin and lipid-lowering medications. These treatments are known to elevate HDL levels and reduce TG levels, accounting for the observed uniformity in the TG/HDL ratio across groups. Moreover, statins may exert protective effects against oxidative stress by upregulating specific antioxidant mechanisms. Statins significantly increased the concentrations of GPx (Zinellu A et al. A Systematic Review and Meta-Analysis of the Effect of Statins on Glutathione Peroxidase, Superoxide Dismutase, and Catalase. Antioxidants (Basel). 2021 Nov 19;10(11):1841. doi: 10.3390/antiox10111841). We add this to discussion (Lines 333-337) . However, according to your suggestion, this correlation remains uncertain, and we agree with you. We have revised this part of the discussion to state: "Surprisingly, a positive correlation emerged between GPx3 levels and the TG/HDL ratio in the prediabetes group. The significance of this correlation remains uncertain and warrants further investigation." (lines 371-373).

Comment: Overall, the study failed to demonstrate a significant correlation that logically links GPx3 and miR196a with metabolic parameters of CMD in prediabetic and T2DM populations for their use as early diagnostic markers. A better population size with a significant statistical correlation is required to conclude the aim.  

Response: Thank you for your feedback on our study. Indeed, we failed to demonstrate that miR-196 can be used as an early biomarker of CMD. However, in the case of GPX3, our results showed that GPX3 can serve as a biomarker of CMD (not as an early biomarker), especially in the T2DM group. This finding was at least partially supported by correlations between GPX3 levels and anthropometric parameters: in the CMD group, weak negative correlations were observed between the level of GPX3 and the mass of adipose tissue, muscle, and total body water (lines 222-223).

Regarding metabolic parameters: "In the CMD group, a weak negative correlation was found between glucose levels and GPX3. Additionally, when analyzing the prediabetes and T2DM groups, this correlation was stronger in the prediabetes group and absent in the T2DM group. This suggests that a significant decrease in GPX3 levels in plasma occurs during the period preceding the development of full-blown T2DM, although such a suggestion was not confirmed by the GPX3 levels in the studied groups (Figures 1a and 1b)." (lines 228-234).

Comments on the Quality of English Language: Minor editing of the English language is required.

Response: We have made every effort to improve the quality of the English language. Additionally, we have utilized the services of the editorial language office.

Reviewer 2 Report

Comments and Suggestions for Authors

Title : Association of glutathione peroxidase 3 (GPx3) and miR-196a 2 with carbohydrate metabolism disorders related 3 anthropometric and metabolic parameters

Reviewer’s comment : The research conducted by the authors is very novel in that it attempted new research using mi RNA on GPX3 and diabetes, which have recently received attention. However, I would like to make some suggestions to improve the completeness of this study in terms of logical development and research design.

Title:  It would be good to mention human experiment research. Expressing a clear research topic and research object will help readers understand the research at a glance.

Abstract:  Please write the full term for GPx3.

Introduction:

-        Please check full term notation and abbreviation rules throughout the manuscript. For example, BMI, WHR, SOD, CAT, etc. must also be modified.

-        The current logical development of the introductory part is somewhat disjointed and overall very plain. It seems that the author needs to use logic to arouse the reader's curiosity, explain the author's field of research that satisfies that curiosity, and appeal to the attractiveness of his or her research.

-        The author explains that the representative marker GPx3 is expressed in the kidneys and lungs, but in fact only measured the concentration of serum circulatory GPx3. Logically explain the validity of the marker set by the author.

Line 80: Please change vitamin c, e to vitamin C, E.

Line 79-80: Please follow the abbreviation rules and correct them.

Line 54-56 : There is a need to provide clearer and more accurate evidence regarding the etiology of IR. When it comes to adipose tissue failure, it is important to explain exactly what the failure is about and clearly present the mechanism by which it is related to IR.

Lines 93-94 Clear reference notation is required.

Lines 99-101 : Clear reference notation is required. Please change GPx-3 to GPx3.

Line 164 : 50% of the participants in the con- Please be careful not to include numbers at the beginning of the sentence.

Manuscript : Authors need to pay close attention to marker expression throughout the manuscript. GPx3 concentration and GPx3 level are used interchangeably. GPx3 activity is also mentioned in the discussion section. This can cause confusion to readers, so it would be better to uniformly express the terminology for markers, provide clear definitions, explain the differences, and logically explain the justification for setting markers.

- When expressing statistical significance, please express P in lowercase italics in P value. There is a need for uniformity throughout.

Line 337: If the phrase “In a current study,” refers to the author’s research, it is appropriate to change it to the definite article the.

Discussion: As the author mentioned, it is essential to measure gene expression rather than explaining the mechanism by setting only the concentration of GPx3 in the blood as a marker. This acts as a significant disadvantage and serves as a basis for weakening the author's argument. Additional experiments and data supplementation are needed for the author's research to have stronger logic. As you know, it is already well known that circulating GPX3 belongs to a cluster of adipokines which is closely related to insulin sensitivity, hyperglycemia, and lipid metabolism(Ref: Identification of adipokine clusters related to parameters of fat mass, insulin sensitivity and inflammation. PLoS One. 2014 Jun;9((6)):e99785.).  Therefore, in this study, the authors need to investigate the correlation between adipokines and Gpx3 in the blood among human subject group.

Comments on the Quality of English Language

There are no problems with English grammar at all, so I think you just need to correct the minor things pointed out.

Author Response

Response to Reviewer 2:

Comment: The research conducted by the authors is very novel in that it attempted new research using miRNA on GPX3 and diabetes, which have recently received attention. However, I would like to make some suggestions to improve the completeness of this study in terms of logical development and research design.

Response: Thank you for recognizing the novelty of our research on the relationship between miR-196a, GPx3, and carbohydrate metabolism disorders. We appreciate your constructive feedback aimed at enhancing the logical development and research design of our study. We are committed to improving our manuscript and would be grateful for any specific suggestions you may have regarding areas that require further attention or elaboration.

Comment: Title:  It would be good to mention human experiment research. Expressing a clear research topic and research object will help readers understand the research at a glance.

Response: Thank you indeed. We propose to revise it to: "Association of Glutathione Peroxidase 3 (GPx3) and miR-196a with Carbohydrate Metabolism Disorders in Elderly”

Comment: Abstract:  Please write the full term for GPx3.

Response: Thank you for pointing out the omission. We will spell out "Glutathione Peroxidase 3" in full at its first mention in the abstract to ensure clarity.

Comment: Introduction:

-        Please check full term notation and abbreviation rules throughout the manuscript. For example, BMI, WHR, SOD, CAT, etc. must also be modified.

Response: Thank you for highlighting the need for consistent notation and abbreviation. We will ensure that terms like BMI, WHR, SOD, and CAT are defined at their first appearance to enhance clarity.

Comment: Introduction:

        The current logical development of the introductory part is somewhat disjointed and overall very plain. It seems that the author needs to use logic to arouse the reader's curiosity, explain the author's field of research that satisfies that curiosity, and appeal to the attractiveness of his or her research.

Response: We appreciate your suggestions on how to improve the logical flow and engagement of our manuscript. To increase transparency, we have added some important information about the risk of CMD in the age group of 65+, the impact of age on antioxidant capacity, and changes in the tone of the discussion regarding the regulation of GPx3 via miR-196. You can find these changes in the Introduction, Results, and Discussion sections. In response, we have made extensive revisions throughout the paper, which are highlighted in red (Abstract: lines 30-32; 35-38; Introduction: lines 52-57, 67-72, 77-80, 86-88, 91-100, 104-124; Discussion: lines 317-322, 335-340, 352-362, 374-375, 378-381, 389-393, 413-416; and Conclusion). Additionally, we have streamlined the Introduction and Discussion sections by removing extraneous information. We have changed our tone in the discussion section where we claim that: “The results of our study suggest that reduced GPx3 levels may serve as a prognostic marker for CMDs in people over 65 years of age. The ROC analysis demonstrated that the risk of CMDs in the group with a GPx3 level of less than 419.501 ng/ml is 5.2 times higher than that in the group with a GPx3 level higher than 419.501 ng/ml (CI:2.31 - 11.58, respectively; OR = 5.175). However, despite initial enthusiasm, this observation is primarily attributable to the significantly lower GPx3 levels found in the T2DM group. In contrast, the reduction in GPx3 in the prediabetes group did not reach statistical significance when compared to the control group. Therefore, the lack of a clear decline in the GPx3 levels in people with prediabetes suggests that GPx3 may not be a reliable early indicator of the CMDs. Although it cannot be ruled out that a larger sample size might have achieved statistical significance, the current data do not support the use of GPx3 as an early predictor of CMD. However, GPx3 may still hold potential as an indicator of progression to diabetes.” (lines: 310-322)

Comment:  Introduction:

 The author explains that the representative marker GPx3 is expressed in the kidneys and lungs, but in fact only measured the concentration of serum circulatory GPx3. Logically explain the validity of the marker set by the author.

Response: Thank you for your suggestion. GPx3, although expressed primarily in the kidney and lung, is also released into the circulation where it can be quantified from serum samples. This approach allows for non-invasive monitoring of GPx3 levels, providing a practical and ethical advantage in human studies. We explained in our manuscript : “Among these isoforms, GPx3 although expressed primarily in the kidney and lung, is also released into the circulation where it can be quantified from serum samples. This approach allows for non-invasive monitoring of GPx3 levels, providing a practical and ethical advantage in human studies.” (Lines 77-80)

Comment:  Line 79-80: Please follow the abbreviation rules and correct them.

Response: We will correct and ensure adherence to standard abbreviation practices throughout the manuscript.

Comment:  Line 54-56 : There is a need to provide clearer and more accurate evidence regarding the etiology of IR. When it comes to adipose tissue failure, it is important to explain exactly what the failure is about and clearly present the mechanism by which it is related to IR.

Response:  Thank you for your suggestion. We have added detailed information about the interplay between insulin resistance, oxidative stress (OxS), hyperglycemia (HG), and aging, which we hope will enhance the clarity of our introduction. The revised section reads:  “First, since 1956, the oxidative theory of aging has become influential [5]. Age-related oxidative stress (OxS) arises from increased free radical production, reduced antioxidants, impaired antioxidant enzymes, and compromised repair mechanisms [6]. Secondly, CMDs form a mutually reinforcing vicious circle. IR results in insufficient cellular response to insulin, causing hyperglycemia (HG). Both high glucose and insulin levels exacerbate oxidative stress (OxS), amplified by aging [6–8]. HI and HG boost nicotinamide adenine dinucleotide phosphate (NADPH) oxidase activity and excessive mitochondrial glucose metabolism triggers an augmented electron flux through the electron transport chain, resulting in a surge in superoxide anion production, which accelerates other reactive oxygen species (ROS) generation [9–12]. These ROS disrupt insulin signaling via the phosphatidylinositol 3-kinase (PI3K/Akt) pathway, induce pancreatic β-cell dysfunction and activating inflammatory pathways [13]. Prolonged HG results in inefficient antioxidant defense, leading to oxidative damage precipitated by ROS and OxS, the primary causes of diabetic complications [14,15]. Elderly people are more vulnerable to OxS due to weakened endogenous antioxidant systems [16]. Aging increases nuclear factor-kappa B (NF-κB) activity via pathways such as mitogen-activated protein kinases (MAPKs) and IκB kinase (IKK), thereby fostering chronic inflammation and impairing the insulin signaling pathway [17]. Consequently, these processes contributes significantly to the initiation and progression of chronic diseases, including CMDs.”  (lines 54-72)

Comment:  Lines 93-94 Clear reference notation is required.

Response: Indeed, references is in the line 93.

Comment:  Lines 99-101 : Clear reference notation is required. Please change GPx-3 to GPx3.

Response: Thank you for highlighting the notation inconsistency for GPx3. We have corrected it and throughout the manuscript to ensure uniformity

Comment:  Line 164 : 50% of the participants in the control- Please be careful not to include numbers at the beginning of the sentence.

Response: Thank you for your attention to the grammatical detail regarding the placement of numbers in our text. We have revised the sentence to read: " In the control group, 50% of the participants were prescribed statins." (Lines 171-172)

Comment:  Manuscript : Authors need to pay close attention to marker expression throughout the manuscript. GPx3 concentration and GPx3 level are used interchangeably. GPx3 activity is also mentioned in the discussion section. This can cause confusion to readers, so it would be better to uniformly express the terminology for markers, provide clear definitions, explain the differences, and logically explain the justification for setting markers.

Response: Thank you for pointing out the inconsistencies in the terminology used for GPx3 in our manuscript. To address this issue, we will standardize our language throughout the manuscript by using “GPx3 level” to refer to its measurable amount in samples. in the original version of our manuscript We discussed GPx3 activity in the discussion section specifically to emphasize that most scientific publications examine GPx3 activity, not its level, which highlights the novelty of our study and its simplicity for easier use in clinical practice. We have removed some information about Gpx3 activity from the discussion, now this information appears only at the end of the discussion and reads: “Our study has limitations, including the lack of cellular GPx activity measure-ments in our patients. Previous analyzes have shown a positive correlation between oxidative stress and cellular GPx activity in individuals with T2DM [63].”( Lines 408-411)

Comment:  When expressing statistical significance, please express P in lowercase italics in P value. There is a need for uniformity throughout.

Response:  Thank you very much for this attention. In agreement, we made the changes in the manuscript.

Comment:  Line 337: If the phrase “In a current study,” refers to the author’s research, it is appropriate to change it to the definite article the.

Response: Thank you very much. We made the changes in the manuscript.

Comment:  Discussion: As the author mentioned, it is essential to measure gene expression rather than explaining the mechanism by setting only the concentration of GPx3 in the blood as a marker. This acts as a significant disadvantage and serves as a basis for weakening the author's argument. Additional experiments and data supplementation are needed for the author's research to have stronger logic. As you know, it is already well known that circulating GPX3 belongs to a cluster of adipokines which is closely related to insulin sensitivity, hyperglycemia, and lipid metabolism(Ref: Identification of adipokine clusters related to parameters of fat mass, insulin sensitivity and inflammation. PLoS One. 2014 Jun;9((6)):e99785.).  Therefore, in this study, the authors need to investigate the correlation between adipokines and Gpx3 in the blood among human subject group.

Response: Thank you for your comment. We recognize the value of your suggestions and agree that exploring a cluster of adipokines would enhance the comprehensiveness of the research in this field. However, we are unable to perform additional experiments not only due to the additional costs but also because all of the original blood samples have been consumed. Considering your valid remark, we have reviewed the literature and discussed other mechanisms that could affect GPX3 levels, which we have added to the discussion section:

"It should be recognized that mechanisms other than miR-196a might affect GPx3 levels. Firstly, histone acetylation or methylation, particularly under the influence of hypoxia—a condition prevalent in excess adipose tissue—increases genome-wide bivalent epigenetic marking, specifically enhancing H3K27me3 levels, which can impact gene regulation [48]. Secondly, GPx3 protein levels may be influenced by various post-translational modifications, including oxidative modifications of cysteine, phosphorylation, acetylation, ubiquitination, glycosylation, protein fragmentation, and protein misfolding [49]. Notably, glycosylation is the most common modification observed in individuals with T2DM [50]." (lines 352-360)

In addition, to address your comment, we indicated the necessity of further research involving adipokine clusters related to parameters of fat mass, insulin sensitivity, and inflammation to better understand the relationship between GPX3 and insulin sensitivity, hyperglycemia, and lipid metabolism. We added this part to the discussion section:

“Hauffe et al. identified diminished GPx3 levels in T2DM patients, which correlated with reduced insulin receptor expression and insulin sensitivity in adipose tissue in multiple mouse model [21]. Flehmig et al. have revealed that GPx3 is a member of a cluster of adipokines, which is closely related to insulin sensitivity, hyperglycemia, lipid metabolism, and inflammation in humans [44]. Additionally, the GPX3 promoter features binding sites for regulatory elements like Sp-1 (specificity protein 1) and HIF-1 (hypoxia-inducible factor-1) as well as a metal response element (MRE) and an antioxidant response element (ARE) [20,45]. Therefore, further research in this field is required.” (Lines 327-335)

Comment:  There are no problems with English grammar at all, so I think you just need to correct the minor things pointed out.

Response: We would like to thank the Reviewer for his opinion.  We have diligently worked to enhance the quality of the English used in our manuscript and have also utilized the editorial office's language editing services to ensure clarity and precision in our presentation.

Round 2

Reviewer 1 Report

Comments and Suggestions for Authors

The authors have addressed the concerns and revised the manuscript reasonably. 

Reviewer 2 Report

Comments and Suggestions for Authors

This manuscript has the advantage of dealing with human samples. The research method has the advantage of securing abundant blood samples. Unfortunately, ELISA kits do not measure the correlation between adipokines and GPx3 in blood samples. However, the logic of the research that was lacking in the discussion section was well supplemented. I hope this will be addressed in future research.

Comments on the Quality of English Language

The sentences were much more organized than the first manuscript. Many of the points pointed out in the manuscript have been corrected.